# Evaluation of Yogurt Quality during Storage by Fluorescence Spectroscopy

**Haifeng Sun [1], Ling Wang [1], Hao Zhang [1,*], Ang Wu [1], Juanhua Zhu [1], Wei Zhang [1] and Jiandong Hu [1,2]**

[1] College of Mechanical and Electrical Engineering, Henan Agricultural University, Zhengzhou 450002, China; haifengdreams@sina.cn (H.S.); wangling0351@126.com (L.W.); cwuang@163.com (A.W.); zhujh88@sina.com (J.Z.); zw@henau.edu.cn (W.Z.); jiandonghu@163.com (J.H.)

[2] State Key Laboratory of Wheat and Maize Crop Science, Zhengzhou 45002, China

[*] Correspondence: hao.zhang2016@hotmail.com; Tel.: +86-0371-6355-8040



**Featured Application: This work provides a potential for the evaluation of yogurt quality during storage by fluorescence spectroscopy.**

**Abstract:** The physico-chemical parameters including pH and viscosity, and the fluorescence signal induced by fluorescent compounds presenting in yogurts such as riboflavin and porphyrin were measured during one week's storage at room temperature when five brands of yogurt samples were exposed to ambient air. The fluorescence spectra of yogurt showed four evident emission peaks, 525 nm, 633 nm, 661 nm, and 672 nm. To quantitatively investigate the quality of yogurt during deteriorating, a calculating method of the average rate of change (ARC) was proposed to study the relative change of fluorescence intensity in the spectral range of 600 to 750 nm associated with porphyrin and chlorin compounds. During the storage, the time evolution of two ARC, pH value, and viscosity were regular. Moreover, the ARC showed a good linear relationship with pH value and viscosity of yogurt. Further, multiple linear regression (MLR) models using two ARC as independent variables were developed to verify the dependence of fluorescence signal with pH value and viscosity, which showed a good linear relationship with an R-square of more than 85% for each class of yogurt. The results demonstrate that fluorescence spectra have a great potential to predict the quality of yogurt.

**Keywords:** fluorescence spectroscopy; yogurt quality; deteriorating; pH; viscosity

---

## 1. Introduction

Yogurt, as one kind of dairy products, has attracted more and more attention in recent years, especially since various brands of yogurt now contain special strains of "probiotics" that can help boost the immune system and promote a healthy digestive tract. Yogurt provides not only rich nutrient elements from milk but also varieties of vitamins produced in the process of bacterial fermentation of milk, which can contribute to health care and prevent diseases [1]. However, when exposed to ambient air, yogurt is vulnerable to microbial contamination and, thus, spoiled. In addition, with the fast development of the yogurt industry, various types of yogurt emerge in the market, followed by the difference in yogurt quality and flavor. Therefore, the accurate identification of different yogurt species and the quantitative evaluation of yogurt quality become very important for the development of the dairy industry.

The traditional detection methods for yogurt quality include biological and chemical methods, such as the standard plate colony counting method for bacterial plate counts and chromatography-mass

spectrometry for protein structure analysis [2–5]. However, these methods are time-consuming, labor-intensive, cumbersome, and destructive. Optical spectroscopy techniques have been widely used to study dairy products due to their advantages of being rapid, having high sensitivity, and being non-invasive. Near-infrared spectroscopy (NIR) technique allows rapid and accurate determination of typical chemical structures presenting in nutrients, such as C–H, N–H, and O–H, due to their characteristic absorption spectrum in the near-infrared range (750–2500 nm) caused by the molecular transition [6–10]. Based on visible/NIR spectroscopy, principal component analysis (PCA) and artificial neural network (ANN) has been used for the discrimination of five kinds of yogurt, as well as for the determination of the sugar content and acidity of yogurt [11,12]. Ultraviolet (UV)-visible spectroscopy is a rapid and alternative technique that provides chemical information, which has been used to monitor the stability of yogurt samples stored at 4 °C up to 49 days [13]. Laser-induced fluorescence (LIF) spectroscopy is a well-known noninvasive method for highly sensitive and selective analysis of molecules, which has been used for the qualitative estimation of proteins in milk at different milking times and for the freshness detection of milk, as well as for studies on deterioration of fresh milk [14–16]. Spatially resolved diffuse reflectance spectroscopy has been used to non-invasively evaluate the fat content in milk and yogurt by measuring the reduced scattering and absorption properties [17]. Although several studies have been conducted, most of which were focused on the quantitative analysis of nutritional components in milk, to our knowledge, few studies have been developed to address the changes of the fluorescence signal and the physico-chemical parameters (such as pH value and viscosity) during yogurt deterioration.

The basic aim of this work was to explore the potential for fast and accurate evaluation of yogurt quality by measuring the fluorescence spectra, pH value, and viscosity during storage. A total of five typical brands of yogurt were investigated. Linear discrimination analysis (LDA) was first attempted to distinguish different yogurt species. To quantify the fluorescence spectra related to porphyrin and chlorin compounds, a calculating method of the average rate of change (ARC) was proposed. Subsequently, the time-dependent ARC, pH value, and viscosity in the process of yogurt storage were analyzed. In the end, multiple linear regression (MLR) models were built to evaluate the relationship between the fluorescence signal with the physico-chemical parameters pH value and viscosity.

## 2. Materials and Methods

### 2.1. Yogurt Samples

Five typical brands of yogurt in China indicated with the same manufacture date and the same shelf life were purchased at a local super-market in Zhengzhou, China. The five brands are HuaHuaNiu (from Zhengzhou, China), JunLeBao (from Shijiazhuang, China), MengNiu (from Neimenggu, China), YiLi (from Neimenggu, China), and GuangMing (from Shanghai, China), which are named by classes A, B, C, D, and E, respectively. The ingredients in yogurt mainly consist of raw milk (≥80%), food additives, and lactobacillus. For each type of yogurt, 20 samples were selected for fluorescence spectral measurements. A total of 100 samples were obtained and placed in glass cups. For the quality measurement of yogurt during storage, all samples were stored in a compartment with the temperature maintained at 23 °C by means of an air conditioner.

### 2.2. Experimental Setup

A typical LIF spectrum measurement system was constructed with a diode laser, a high-pass filter, three biconvex lenses, a multi-mode fiber, and a fiber-optical spectrometer. The diode laser with an emission wavelength of 405 nm and an output power of 50 mW shot the light onto the surface of the yogurt sample by a focusing lens. The excited fluorescence was collected by an optical unit consisting of a high-pass filter as well as two focusing lenses with the same focusing lens. To decrease the directly reflected light from the surface of the yogurt sample, the optical unit was arranged with a 45-degree angle. After that, the scattered excitation light was further suppressed by using a high-pass

filter with the cutoff wavelength of 420 nm. The sample fluorescence was focused into the port of a multi-mode optical fiber with a core diameter of 600 μm and then transported to a portable spectrometer (USB2000+, Ocean Optics, USA). In the end, the spectral data were stored by a personal computer (PC) for further analysis.

### 2.3. pH and Viscosity Measurements

The pH value and viscosity of the yogurt samples were measured using a pH-meter (pH-100, Lichen Instruments, Shanghai, China) and a viscometer (LND-1, Lichen Instruments, Shanghai, China), in the condition of room temperature. Before measurements, the pH-meter was calibrated using standard buffer solutions of pH = 4.00, pH = 6.86, and pH = 9.18 at room temperature (about 25 °C). After each measurement, the pH-meter and viscometer were washed with ultra-pure water. For each sample, three independent measurements were performed to obtain the average value.

### 2.4. Statistical Analysis

In this work, linear discriminant analysis (LDA) based on the principal component analysis (PCA) was firstly used for identifying five species of yogurt. Principal component analysis (PCA) is an unsupervised dimension-descending method, which reduces the dimensions of the original spectral data matrix with the minimal loss of information by decomposing the data matrix into a structure part and a noise part. The decomposing process is expressed by [18]

$$X = TP^T + E \tag{1}$$

where $P$ is the transformation matrix (or loading matrix) between the original variable space and the new data space decided by the principal components (PCs). The loading vectors in the matrix $P$ are the representations of the principal components in the original variables. $T$ is the score coefficient matrix, which are the coordinates of all data points in the new principal component space. $E$ represents the residual noise.

The principle of LDA is to find out a so-called discriminant function which best separates the classes by minimizing the distance of within-class samples and maximizing the distance of between-class samples. The LDA is also a projection method by projecting the data into a multidimensional straight line and then selecting an appropriate line to separate the different data points. The projection function is given as:

$$y_j = w_j^T x + w_{j0} \tag{2}$$

where, $x \in X_j$, $X_j$ is the $j$th sample set. $j = 1, 2, \ldots, k$, $k$ is the class number of the total samples. Then the discriminant function is given by [19]:

$$J(W) = \frac{\prod\limits_{diag} W^T S_b W}{\prod\limits_{diag} W^T S_w W} = \prod_{i=1}^{d} \frac{w_i^T S_b w_i}{w_i^T S_w w_i} \tag{3}$$

where $W$ is the projection matrix composing of the maximum eigenvalue of the matrix, $d$ is the number of the maximum eigenvalue. $S_w$ and $S_b$ are the within-class scatter matrix, and the intra-class scatter matrix, respectively, which are expressed by:

$$S_b = \sum_{j=1}^{k} N_j (\mu_j - \mu)(\mu_j - \mu)^T \tag{4}$$

where $\mu$ is the mean value of the total samples, $\mu_j$ is the mean value of the $j$th class samples, $N_j$ is the number of the $j$-class samples.

In this study, after performing PCA to reduce the dimensionality of variables, LDA was carried out to develop a linear discrimination model by using the score values of the first six PCs that gave more than 99.9% explained percentage.

## 3. Results and Discussion

### 3.1. Spectral Investigation

Fluorescence spectra were obtained in the wavelength range of 339.98 nm to 1028.70 nm with a resolution of 0.38 nm. Each spectrum includes 2048 data points which are decided by the linear CCD array of USB 2000+ spectrometer. To reduce the number of data points available for multivariate analysis, the spectra in the wavelength range of 410 to 910 nm, where all available fluorescence signals are included, was selected. The average fluorescence spectra measured from five different kinds of yogurt are shown in Figure 1. As can be observed clearly, there are several spectral wavelength bands that might be particularly interesting. The prominent emission band with a peak wavelength of 525 nm located in the wavelength range of 500 to 600 nm has almost the same spectral shape for five kinds of yogurt, which means that the emission spectra in this range should be attributed by the fluorophores presenting in raw milk, such as tryptophan, nicotinamide adenine dinucleotide (NADH), vitamin A, riboflavin, and Maillard products [20,21]. Another interesting region is from 600 to 750 nm, where five different narrow emission bands occur. The first emission band appears approximately at 633 nm, which has a similar spectral shape for different kinds of yogurt samples. An apparent double peak appearing at about 661 nm and 672 nm have a slight difference in the spectral intensity, for example, the peak intensity at 661 nm of class D is far less than that at 672 nm. These narrow emission peaks in the red and near-infrared (NIR) region might be produced by the presence of porphyrin and chlorin compounds in yogurt, most likely protoporphyrin, hematoporphyrin, chlorophyll a and chlorophyll b [22,23]. In addition, a small peak appearing at about 420 nm is the same for both the spectral shape and intensity for the five kinds of yogurt, which should be attributed to the fluorescence signal from the high-pass filter when excited with 405 nm laser so this peak can be ignored in the present study.

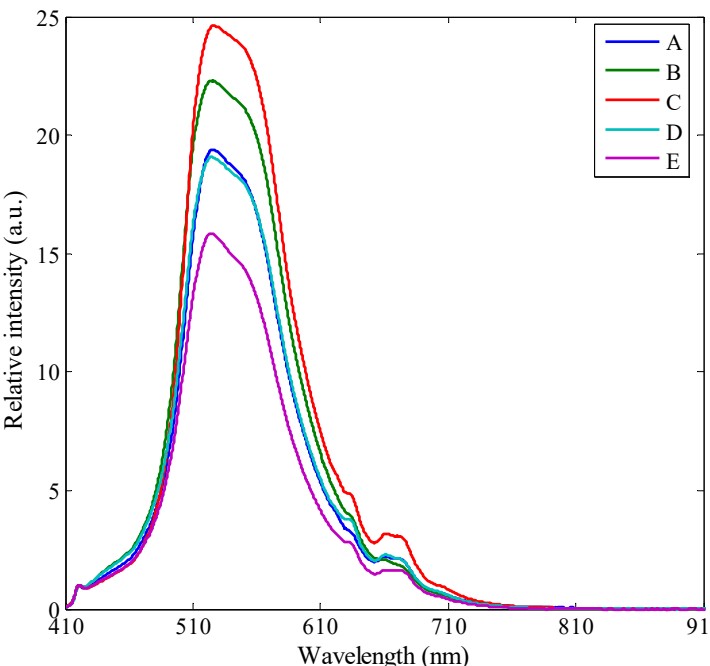

**Figure 1.** Average fluorescence spectra of five brands of yogurt (HuaHuaNiu, JunLeBao, MengNiu, YiLi, and GuangMing).

### 3.2. Yogurt Classification Based on LDA

Based on the PCA score coefficient matrix, LDA was used to improve the classification accuracy between different brands of yogurt that were difficult to separate using only PCA. Since the number of PCs selected for the LDA model will influence the classification result, the first six PCs accounting for over 99.99% of the total variance of the data matrix were chosen as the optimal variables to develop the LDA discrimination model. Thus, the $100 \times 1459$ data matrix of fluorescence spectra was transformed to a new dataset consisting of a $100 \times 6$ data matrix. To establish a suitable LDA discrimination model and then evaluate the model, the new dataset was divided into a training set and a prediction set. Each set consists of 50 fluorescence spectra (a $50 \times 6$ data matrix), where the training set was used to establish the LDA discrimination model, and the prediction set was used to predict the feasibility of discrimination model.

Figure 2a depicts the discrimination results of different yogurt brands by means of 2D scatter plot of the first two most important discriminant functions, an 85% confidence ellipse was used to describe the accuracy of predicted discrimination. As can be obserrved clearly, the yogurt samples are reasonably well discriminated. The samples of the classes A, B, and D show a good separation, while the samples of the classes C and E have several overlaps, which might produce some misclassifications.

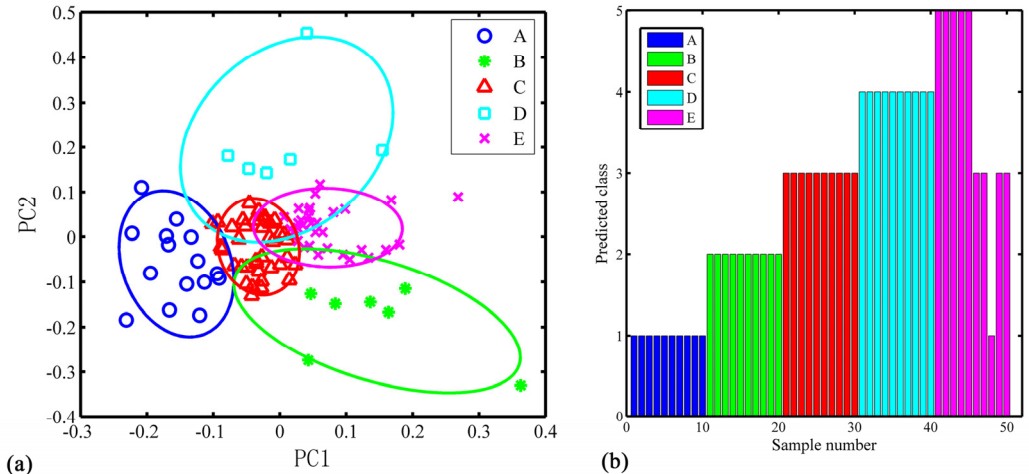

**Figure 2.** (**a**) Linear discriminant analysis for five different brands of yogurt samples (A, B, C, D, and E) based on the first two discriminant functions; (**b**) Bar plot of LDA discrimination model for the predication set from five different kinds of yogurt.

The predicted classification result of the LDA model is shown in Figure 2b, where the classes A, B, C, D, and E are indicated by the color of blue, green, red, cyan, and mauve, respectively. It can be observed clearly that almost all the fluorescence spectra of yogurt samples (47 samples) except five samples from class E in the prediction set were correctly discriminated. For these five misclassified samples from class E, one was identified as the class A, and the other four were identified as the class C. Therefore, the classes A, B, and D have a good discrimination, the misclassification of the classes C and E are mostly caused by the overlap of discriminant functions shown in Figure 2a. The results demonstrate the LDA discrimination model can be successfully employed for the accurate classification of yogurt brands.

### 3.3. The Quality Evaluation

Each sample was regularly measured over a period of one week to monitor the deterioration process of yogurt samples, measurements were performed at a regular time every day since the porphyrins and chlorins naturally existing in yogurt both can be acted as photosensitizers, which are highly sensitive to light and easy to suffer from light-induced degradation. By selecting the fluorescence spectral band in the wavelength range of 600 to 650 nm, where five narrow shape emission peaks

attributed to the porphyrins and chlorins are presented, the normalized fluorescence spectra of five brands of yogurt are shown in Figure 3A–E. During the successive measurements from one to seven days, these emission peaks exhibit major changes due to the photodegradation of the porphyrins and chlorins. Therefore, these fluorescence peaks can be used as indicators to estimate the changes of the porphyrins and chlorins during yogurt deterioration.

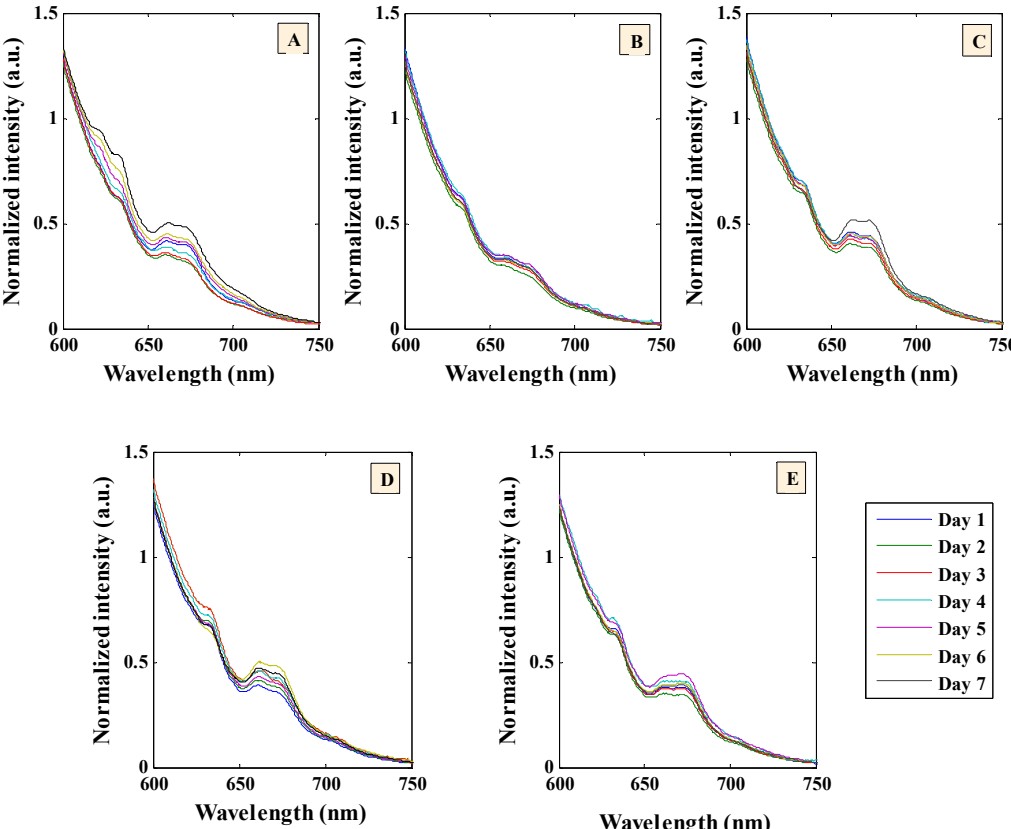

**Figure 3.** Normalized fluorescence spectra of each kind of yogurt during the measurements over one week.

To extract the spectroscopic information related to time-dependent deterioration process of yogurt, a calculating method of the average rate of change (ARC) was proposed, which can be used to evaluate the relative change of fluorescence intensity in a given spectral range. The formula for ARC can be expressed by:

$$K_{\lambda_1/\lambda_2} = \frac{\Delta I}{\Delta \lambda} = \frac{I_{\lambda_1} - I_{\lambda_2}}{\lambda_1 - \lambda_2} \tag{5}$$

where $I_{\lambda_1}$ and $I_{\lambda_2}$ are the fluorescence intensity at the wavelength of $\lambda_1$ and $\lambda_2$. Considering the fluorescence change related to porphyrin and chlorin compounds in the wavelength range of 600 to 750 nm, $K_{622/633} = \frac{I_{622} - I_{633}}{622 - 633}$ was defined for estimating the change of fluorescence peak at 622 nm and 633 nm, $K_{661/672} = \frac{I_{661} - I_{672}}{661 - 672}$ was defined for evaluating the peak change at 661nm and 672 nm.

The time evolution of calculated $K_{622/633}$ and $K_{661/672}$ for these five types of yogurt samples are displayed in Figure 4. A decreasing tendency of $K_{622/633}$ and an increasing tendency of $K_{661/672}$ are clearly found during the whole measurements of seven days. As suggested by Wold [22], the peak at 633 nm could be attributed to protoporphyrin, while the double peak at 661 nm and 672 nm might mostly be due to the chlorophyll residues. Therefore, the decrease of $K_{622/633}$ indicates the photodegradation of protophyrin, and the increase of $K_{661/672}$ represents the photodegradation of chlorophyll residues. The results show that the characteristic fluorescence intensity is highly sensitive to the changes of yogurt composition during the deterioration process, and therefore the fluorescence

peaks of photodegradation of protophyrin could be used as indicators to quantitatively estimate the quality of yogurt during storage.

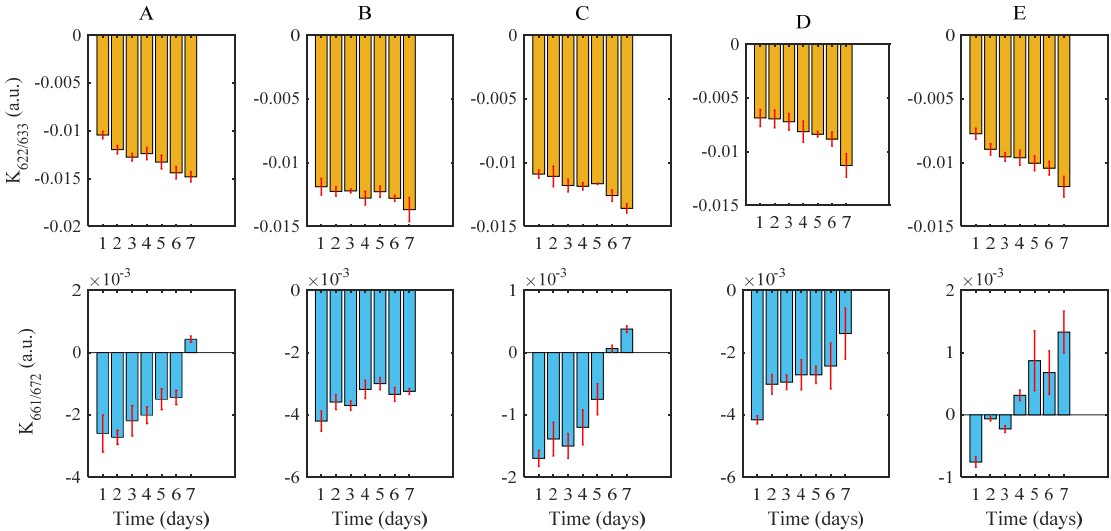

**Figure 4.** $K_{622/633}$ and $K_{661/672}$ values calculated for yogurt samples (A, B, C, D, and E) over one week.

For comparison purposes, the physico-chemical parameters pH value and viscosity of yogurt samples were also measured during the deteriorating process. As shown in Figure 5, a decreasing tendency was found both for pH value and viscosity during a period of 7 days, while individually, each type of yogurt sample has a slight difference. The reduction of pH in yogurt could be mostly caused by the production of lactic acid as a result of the action of the starter bacteria, which was actually apparent in the case of room temperature storage. The pronounced viscosity-decreasing behavior might be attributed to the protein denaturation and the destruction of protein colloid structure in the room temperature. Therefore, the reduction of pH value and viscosity determinate the quality of yogurt.

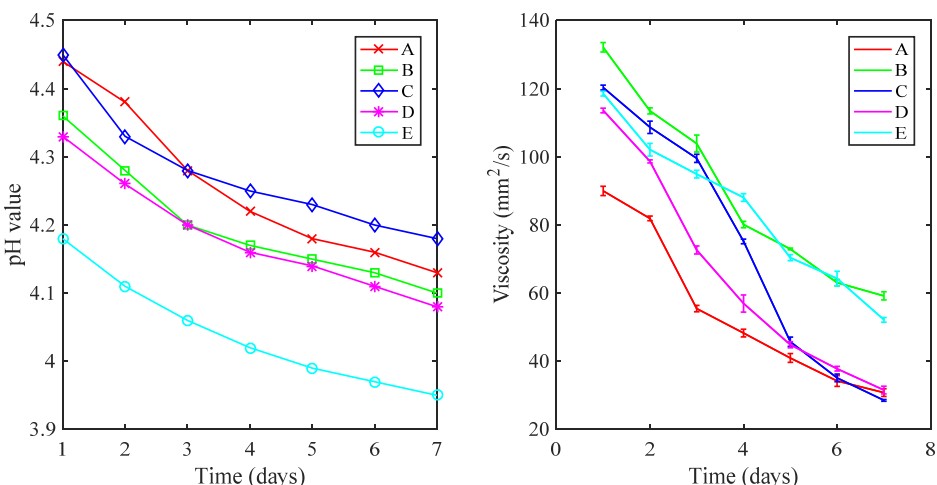

**Figure 5.** pH value and viscosity of yogurt samples (A, B, C, D, and E) measured over one week.

The $K_{622/633}$ value was selected to find out the relevance between fluorescence spectra and the physico-chemical parameters during yogurt deterioration since it had a similar time evolution with pH value and viscosity. As displayed in Figure 6, good linear relationship is achieved both for pH value and viscosity, which indicates that $K_{622/633}$ value shows a strong correlation with the physico-chemical parameters of yogurt. The physico-chemical parameters pH and viscosity are regarded as two most

common indicators for the evaluation of yogurt quality, which further verifies that fluorescence signals can be used as one indicator to quantitatively estimate the quality of yogurt.

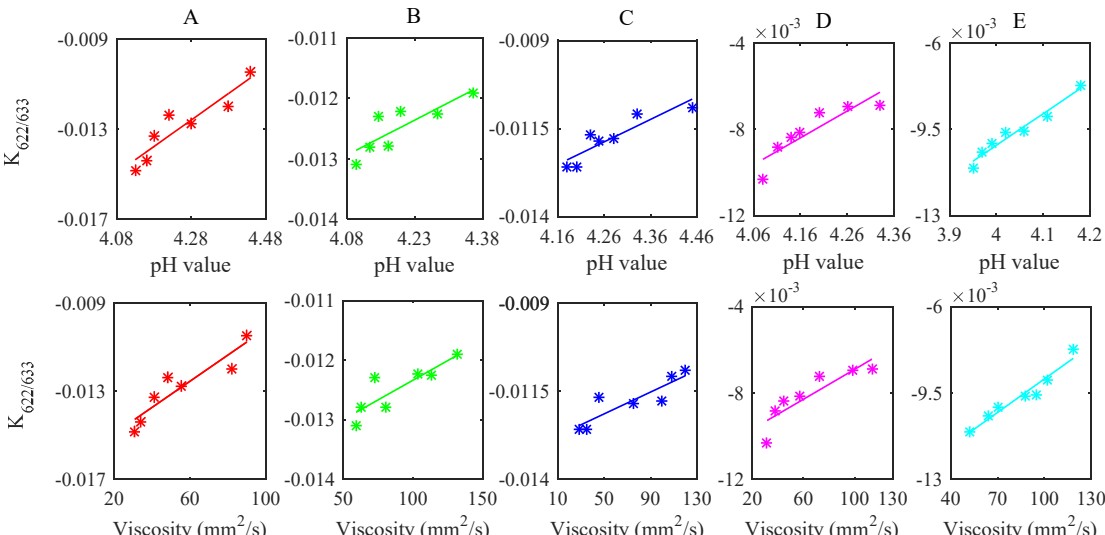

**Figure 6.** The relevance of fluorescence peak ratio $K_{622/633}$ with pH value and viscosity of yogurt during the whole measurement series.

To further validate the dependence of fluorescence signal with pH value and viscosity, multiple linear regression (MLR) models were developed, where $K_{622/633}$ and $K_{661/672}$ value were used as independent variables, pH value and viscosity were selected as dependence variables, respectively. As shown in Figure 7, $K = p1 \cdot K_{622/633} + p2 \cdot K_{661/672}$, where $K$ is the normalized value dependent on both $K_{622/633}$ and $K_{661/672}$ value, $p1$ and $p2$ are the coefficients. Each MLR regression model presents a correlation coefficient ($R^2$) of more than 0.85, where classes B and E have a preferable linear relationship, which again indicates that fluorescence measurements have the potential to estimate the yogurt quality.

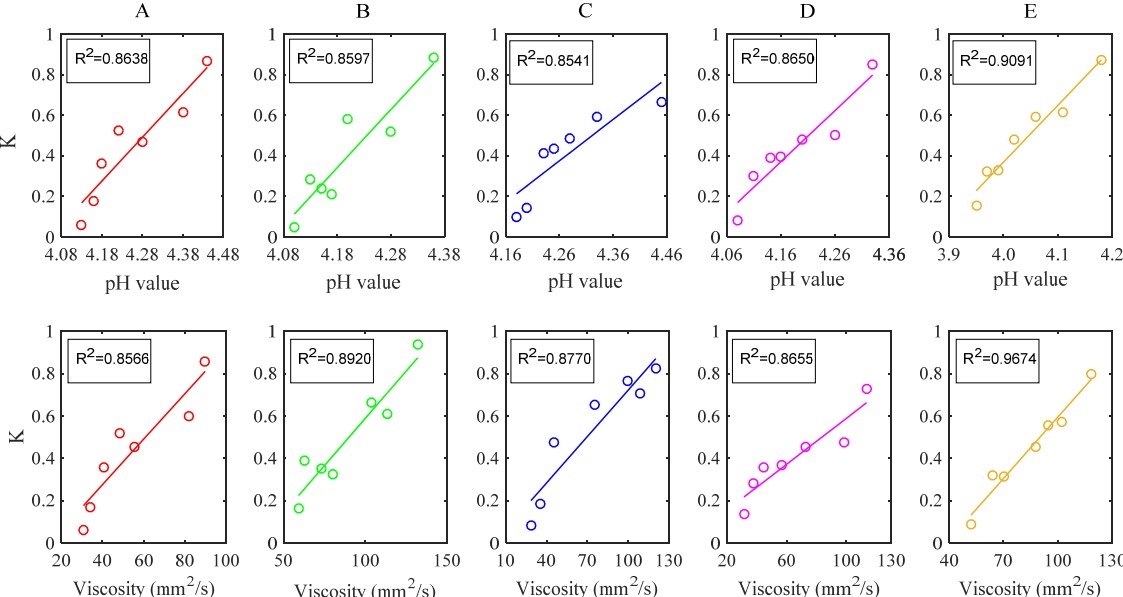

**Figure 7.** Predicted pH value and viscosity of yogurt samples based on MLR model.

## 4. Conclusions

The present work demonstrates that fluorescence spectroscopy can be used rapidly and non-invasively for the evaluation of yogurt quality during deteriorating. The characteristic fluorescence spectra of yogurt consist of a broadband spectrum in the wavelength region of 500 to 600 nm with a peak at 525 nm attributed mainly by the riboflavin and Maillard products of raw milk and several narrow emission peaks in the region of 600 to 750 nm caused by the porphyrin and chlorin compounds existing naturally in yogurt. Based on a total of 100 fluorescence spectra from five brands of yogurt, LDA discrimination model was carried out to classify the yogurt brands, which showed a preferable classification result. To realize the prediction of yogurt quality, two ARC formulas $K_{622/633}$ and $K_{661/672}$ were utilized to extract the spectral information related to yogurt deterioration, particularly in the 600 to 750 nm region. The decreasing tendency of $K_{622/633}$ and the increasing tendency of $K_{661/672}$ indicate the photodegradation of porphyrin and chlorin compounds. Moreover, the fluorescence peak ratio $K_{622/633}$ shows a good linear relationship with the measured pH value and viscosity of yogurt, which further verifies the physico-chemical change related to the quality of yogurt. Based on these two fluorescence peaks ratios $K_{622/633}$ and $K_{661/672}$, MLR models were established to verify the dependence of fluorescence signal with pH value and viscosity. A more than 85% correlation coefficient is obtained for each class of yogurt, which further demonstrates the potential and effectiveness for the evaluation of yogurt quality by using fluorescence spectroscopy.

**Author Contributions:** H.S. and H.Z. performed the fluorescence spectral measurements. L.W. and A.W. performed the measurements of pH value and viscosity. J.Z. and W.Z. were involved in the data processing and data analysis. H.Z. and J.H. were involved in writing and revising the paper.

**Funding:** This work was financially supported by the China Postdoctoral Science Foundation (No. 2017M612399), the Science and Technology Project of Henan Province (No. 182102110427), the Science and Technology Innovation Project of Henan Agricultural University (No. KJCX2018A09), the National Natural Science Foundation of China (No. 31671581), the Natural Science Foundation of Henan Province (No. 162300410143), and the Science and Technology Project of Henan Province (No. 182102110250).

**Conflicts of Interest:** The authors declare no conflict of interest.

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
