# Peer review of "Evaluation of Yogurt Quality during Storage by Fluorescence Spectroscopy"

_applsci, doi:10.3390/app9010131_

Reviewer 1 Report

 Comments to the authors

I carefully reviewed the manuscript entitled: “Quality evaluation of yogurt during deteriorating by fluorescence spectroscopy combined with the physico-chemical parameters”. While I found it to be interesting and innovative, I have a few suggestions and questions for the authors. See list below.

1.     The title of the paper is confusing. Edit English language, for clarity. Maybe change to: Evaluation of yogurt quality during storage by fluorescence spectroscopy?

2.     Replace “PH” for “pH” in all sections of the article

3.     Line 18. What does “species” of yogurt mean? Did the authors mean to say “samples”, “types”,” brands”? Please, clarify

4.     Line 24. The authors describe the evolution of ARC, pH, and viscosity as “regular”. What does that mean?

5.     Line 33. Do all yogurts contain microorganisms considered as “probiotics”?

6.     Line 38. Why would manufacturers expose the yogurt to ambient air? What kind of microbes would normally “contaminate” the product, survive the low pH and outcompete lactic acid bacteria?

7.     Line 46. Did the authors mean to say “dairy products…”? It reads “daily”

8.     Line 84. Materials and methods. Explain why the authors conducted the study at 23 ˚C. What’s the rationale? Seems like a very high temperature for long term storage of yogurt. The experiment would have more real-world application if done at refrigeration temperature.

9.     Line 76. Edit English grammar.

10.  Line 101. Why did the authors calibrate the pH meter using only one standard? It is recommended to use at least 2

11.  Line 259. Conclusions. Looks like the use of fluorescence spectroscopy has potential for the evaluation of the shelf-life of yogurt. But when the authors mention that using that technique combined with pH and viscosity measurements it really looses importance. Why would manufactures use a method that is more expensive in addition to well established, affordable, and simple methodologies such as measuring viscosity and pH? Explain.

Author Response

Point 1:The title of the paper is confusing. Edit English language, for clarity. Maybe change to: Evaluation of yogurt quality during storage by fluorescence spectroscopy?

Response 1: According to Reviewers suggestion, the title of the manuscript has changed to “Evaluation of yogurt quality during storage by fluorescence spectroscopy”

Point 2: Replace “PH” for “pH” in all sections of the article

Response 2: According to Reviewers suggestion, all misspelled PH in the manuscript has been changed to pH.

Point 3: Line 18. What does “species” of yogurt mean? Did the authors mean to say “samples”, “types”,” brands”? Please, clarify

Response 3: Line 18. In the manuscript, yogurt species means different brands of yogurt, we have changed species to brands in the manuscript.

Point 4: Line 24. The authors describe the evolution of ARC, pH, and viscosity as “regular”. What does that mean?

Response 4:  Line 24. In the manuscript,  regular means that the change of ARC, pH, and viscosity over time was orderly, for example, the pH was decreasing over time, and the viscosity was increasing over time.

Point 5:  Line 33. Do all yogurts contain microorganisms considered as “probiotics”?

Response 5: Line 33. Yes, not all yogurt contain probiotics. Indeed, all yogurts contain the bacteria Lactobacillus bulgaricus and Streptococcus thermophilus.  We have changed the sentence Yogurt, as one kind of dairy products, has attracted more and more attentions in recent years, due to the presence of probiotics that can help boost the immune system and promote a healthy digestive tract. to “Yogurt, as one kind of dairy products, has attracted more and more attentions in recent years, especially since various brands of yogurt now contain special strains of “probiotics” that can help boost the immune system and promote a healthy digestive tract. ”

Point 6: Line 38. Why would manufacturers expose the yogurt to ambient air? What kind of microbes would normally “contaminate” the product, survive the low pH and outcompete lactic acid bacteria?

Response 6: Line 38. Thats right. Normally, yogurt should be sealed for manufacturers, while for consumers, the yogurt might be exposed to ambient air when they cannot consume at once, especially for the big barrel of yogurt, so the consumers need to know if the yogurt is edible after a period of time. When exposed to air, the deterioration of yogurt is mainly caused by the contaminate of saccharomycetes and mycete, which have been reported previously [R1, R2].

References

[R1] Niamah, A. K. (2017). Physicochemical and Microbial Characteristics of Yogurt with Added Saccharomyces Boulardii. Current Research in Nutrition and Food Science Journal, 5(3), 300-307.

[R2] Knight-Jones, T. J., Hangombe, M. B., Songe, M. M., Sinkala, Y., & Grace, D. (2016). Microbial contamination and hygiene of fresh cows milk produced by smallholders in Western Zambia. International journal of environmental research and public health, 13(7), 737.

Point 7: Line 46. Did the authors mean to say “dairy products…”? It reads “daily”

Response 7:  Line 46. According to Reviewers suggestion, all misspelled daily in the manuscript has been changed to dairy.

Point 8: Line 84. Materials and methods. Explain why the authors conducted the study at 23 ˚C. What’s the rationale? Seems like a very high temperature for long term storage of yogurt. The experiment would have more real-world application if done at refrigeration temperature.

Response 8: Line 84. The reason that we conducted the study at the room temperature was to observe the deterioration process of yogurt by using fluorescence spectroscopy combined with pH and viscosity measurements when exposed to air. Indeed, the measurement process cannot indicate the real storage time, the real storage time is very short (might one or two days). In our experiments, in vision and  taste, after one day, the yogurt began to solidify. Of course, it is more useful and applicable when the experiment were done at low temperature. We have conducted similar work to measured the yogurt storage time by placing the yogurt at refrigeration temperature using laser spectroscopy, please refer to [R3].

References

[R3] Li, T., Lin, H., Zhang, H., Svanberg, K., & Svanberg, S. (2017). Application of Tunable Diode Laser Spectroscopy for the Assessment of Food Quality. Applied spectroscopy, 71(5), 929-938.

Point 9: Line 76. Edit English grammar.

Response 9: Line 76. the sentence “A total of 5 typical kinds of yogurt made with the same manufacture date and the same indicated shelf life were purchased at a local super-market in Zhengzhou, China.” has changed to “5 typical brands of yogurt in China indicated with the same manufacture date and the same shelf life were purchased at a local super-market in Zhengzhou, China.”

Point 10: Line 101. Why did the authors calibrate the pH meter using only one standard? It is recommended to use at least 2

Response 10: Line 101. Indeed, in the first test, we used three standard buffer solutions to calibrate the pH-meter, i.e. pH=4.00, pH=6.86, and pH=9.18 at the room temperature (about 25 °C). While after several test, we found that the change of pH is very small when we only used pH=6.86 buffer solution, which was almost the same to that calibrated with three buffer solutions. Therefore, in the later experiments, we just selected pH=6.86 buffer solution for calibration. The detail information of these three buffer solutions are list below. The sentence “Before measurements, the pH-meter was calibrated using standard buffer solutions of pH=6.86.” has been changed to “Before measurements, the pH-meter was calibrated using standard buffer solutions of pH=4.00, pH=6.86, and pH=9.18 at the room temperature (about 25 °C).”
Point
11: Line 259. Conclusions. Looks like the use of fluorescence spectroscopy has potential for the evaluation of the shelf-life of yogurt. But when the authors mention that using that technique combined with pH and viscosity measurements it really looses importance. Why would manufactures use a method that is more expensive in addition to well established, affordable, and simple methodologies such as measuring viscosity and pH? Explain.

Response 11: Line 259. Conclusions. Thats right. The aim of our work was to use fluorescence spectroscopy to explore the possibility for the evaluation of the quality of yogurt during storage, while the pH and viscosity measurements was mainly used as complementary tools to further validate the potential of fluorescence spectroscopy. We have revised the sentence in the part of Conclusions, please check it.

Reviewer 2 Report

Please find attached suggestions for improvement of the manuscript. Please check the information regarding to the starter cultures present in the 5 types of yogurt. Yogurt is expected to contain both Lactobacillus bulgaricus and Streptococcus thermophillus.

Author Response

Response : 

According to the Reviewers suggestions, in the revised manuscript, all misspelled PH has been changed to pH, and all misspelled daily has been changed to dairy.

Line 48, compositions has been changed to structures.

Line 49, its has been changed to their;

Line 51, have been has been changed to has been;

Line 58, for the studies on the deteriorating has been changed to “for studies on deterioration of fresh milk”;

Line 60, a mass of has been changed to several;

Line 64, deteriorating has been changed to “deterioration ”;

Line 67, investigated, linear discrimination analysis has been changed to investigated. Linear discrimination analysis;

Line 83-84, all samples were stored in a room temperature, kept at 23 °C by an air conditioner has been changed to all samples were stored in a compartment with the temperature maintained at 23 °C by means of an sir conditioner;

Line 86, were has been changed to was;

Line 101, measurements has been changed to measurement

Line 105-106, principle has been changed to principal, Principle has been changed to Principal;

Line 137, 0.38 nm, each spectrum has been changed to 0.38 nm. Each spectrum;

Line 139-140, the spectra in the wavelength range of 410 - 910 nm were selected, where all available fluorescence signals are included has been changed to the spectra in the wavelength range of 410 - 910 nm, where all fluorescence signals are included, was selected.;

Line 162, only using has been changed to using only;

Line 168, prediction set, each set has been changed to prediction set. Each set;

Line 192, every day. Since the has been changed to every day, since;

Line 202, of one week has been changed to over one week;

Line 230-231, could be mostly caused by the production of lactic acid decomposed by the lactobacillus, which was actually apparent in the case of room temperature storage. has been changed to could be mostly caused by the production of lactic acid as a result of the action of the starter bacteria, which was actually apparent in the case of room temperature storage.;

Line 236, in one week has been changed to over one week;

Line 264, might has been deleted;

Line 274, viscosity, more than has been changed to viscosity. More than;

Line 298, Food Control 2017 has changed to Food Control 2017

Point 1: Line 34, in fact, conventional yogurt does not contain probiotic bacteria.

Please change sentence.

Response 1: Line 34. Yes, thats right, not all yogurt contain probiotics. We have changed the sentence  Yogurt, as one kind of dairy products, has attracted more and more attentions in recent years, due to the presence of probiotics that can help boost the immune system and promote a healthy digestive tract. to “Yogurt, as one kind of dairy products, has attracted more and more attentions in recent years, especially since various brands of yogurt now contain special strains of “probiotics” that can help boost the immune system and promote a healthy digestive tract. ”

Point 2: Line 37. “care and prevent diseases”. This should be supported by references.

Response 2: One reference has been added, i.e. “Astrup, A. Yogurt and dairy product consumption to prevent cardiometabolic diseases: epidemiologic and experimental studies. The American journal of clinical nutrition 2004, 99(5), 1235S-1242S.”

Point 3:  Line 39. “emerge in the market, followed by the appearance of conuterfeit.”, sentence not clear

Response 3: Line 39,  the sentence “various types of yogurt emerge in the market, followed by the appearance of counterfeit.” has been changed to “various types of yogurt emerge in the market, followed by the difference of yogurt quality and flavor.”.

Point 4: Line 158-159. (HuaHuaNiu, JunLeBao, MengNiu, YiLi, and GuangMing)???

Response 4: HuaHuaNiu, JunLeBao, MengNiu, YiLi, and GuangMing indicated the brand name of each type of yogurt, which was named by classes A, B, C, D, and E, respectively. Please refer to Line 78-80 in the manuscript.

Point 5: Line 174-175. “The samples of the classes A, B, and D show a good separation, while the samples of the classes C and E have several overlaps” is that correct? This sample seems to overlap with B,C and E.

Response 5: Sorry for the mistake when we plotted the figure, in Figure 2(a), the order of C and D was wrong. We have provided the new figure, as shown below, which also replaced the old figure in the manuscript.

Point 6: Line 265-267. “Based on the spectral characteristics of yogurt fluorescence, LDA discrimination model was carried out to classify the yogurt species, where a preferable classification results were observed.” sentence not clear

Response 6: Line 265-267,  the sentence Based on the spectral characteristics of yogurt fluorescence, LDA discrimination model was carried out to classify the yogurt species, where a preferable classification results were observed. has been changed to Based on a total of 100 fluorescence spectra from five kinds of yogurt, LDA discrimination model was carried out to classify the yogurt brands, which showed a preferable classification results.
